# Quantitative Analysis of Acetone in Transformer Oil Based on ZnO NPs@Ag NWs SERS Substrates Combined with a Stoichiometric Model

**DOI:** 10.3390/ijms232113633

**Published:** 2022-11-07

**Authors:** Xinyuan Zhang, Yu Lei, Ruimin Song, Weigen Chen, Changding Wang, Ziyi Wang, Zhixian Yin, Fu Wan

**Affiliations:** State Key Laboratory of Power Transmission Equipment & System Security and New Technology, Chongqing University, Chongqing 400044, China

**Keywords:** SERS, ZnO NPs@Ag NWs, acetone, transformer oil, quantification

## Abstract

Acetone is an essential indicator for determining the aging of transformer insulation. Rapid, sensitive, and accurate quantification of acetone in transformer oil is highly significant in assessing the aging of oil-paper insulation systems. In this study, silver nanowires modified with small zinc oxide nanoparticles (ZnO NPs@Ag NWs) were excellent surface-enhanced Raman scattering (SERS) substrates and efficiently and sensitively detected acetone in transformer oil. Stoichiometric models such as multiple linear regression (MLR) models and partial least square regressions (PLS) were investigated to quantify acetone in transformer oil and compared with commonly used univariate linear regressions (ULR). PLS combined with a preprocessing algorithm provided the best prediction model, with a correlation coefficient of 0.998251 for the calibration set, 0.997678 for the predictive set, a root mean square error in the calibration set (RMSECV = 0.12596 mg/g), and a prediction set (RMSEP = 0.11408 mg/g). For an acetone solution of 0.003 mg/g, the mean absolute percentage error (MAPE) was the lowest among the three quantitative models. For a concentration of 7.29 mg/g, the MAPE was 1.60%. This method achieved limits of quantification and detections of 0.003 mg/g and 1 μg/g, respectively. In general, these results suggested that ZnO NPs@Ag NWs as SERS substrates coupled with PLS simply and accurately quantified trace acetone concentrations in transformer oil.

## 1. Introduction

The oil-paper insulation material of transformers and other electrical equipment will be subject to the combined effects of electrical, thermal, mechanical stress, and other factors during the operation process, which will cause the aging and deterioration of the insulation structure, leading to the occurrence of faults and seriously endangering the safe and stable operation of the power system [1,2,3,4,5,6,7,8]. During the aging process, the oil-paper insulation material decomposes and produces acetone, furfural, CO, CO_2_, etc., which are dissolved in the transformer oil. The content of these characteristic aging products can indirectly reflect the degree of aging of the oil-paper insulation of electrical equipment [9,10,11,12,13,14,15,16,17]. As one of the aging characteristics of transformer oil, acetone can not only exist stably in the oil but also accurately evaluate the aging state of the transformer, which makes it ideal for assessing the aging of insulation systems. The concentrations of acetone from 0.012 to 0.1 mg/g were listed as attention values for the aging of oil-paper insulation [18,19,20,21,22,23]. Nowadays, headspace gas chromatography and gas chromatography-mass spectrometry are mainly applied to assess the acetone in transformer oil domestically and overseas. While their sensitivity and reliability have been reported previously, they suffer from a complicated operation, high cost, and long detection time [24,25,26,27,28,29]. Consequently, it is increasingly necessary to find a simple, rapid, and inexpensive method for the detection of acetone in transformer oil. 

Raman spectroscopy, an effective method for the detection and analysis of liquid materials, can be applied to the determination of the types and concentrations of the characteristic substances [30,31,32]. Compared with headspace gas chromatography and gas chromatography-mass spectrometry, Raman spectroscopy is simple in preparing samples with high detection speed, and in addition, a very small sample size is required for detection. In 2018, conventional confocal laser Raman spectroscopy was first used to analyze the detection characteristics of acetone in transformer oil with a minimum detection concentration of 45 mg/L, which is far from the requirement of practical engineering detection [33]. Perhaps surface-enhanced Raman scattering (SERS) is a more powerful tool for the detection of trace acetone based on its molecular signal enhancement and real-time applicability [34,35,36,37]. Colloidal suspensions of precious metals such as Au and Ag nanoparticles are commonly used as active SERS substrates due to their strong localized surface plasmon resonance in the visible region, where strong electromagnetic fields can be generated, which is known as the electromagnetic enhancement (EM) [38,39,40,41]. Silver nanowires (Ag NWs) are unique plasmonic nanomaterials that support propagating surface plasmon excitations (SPPs) because the sharp edges or branches of silver metal nanostructures can aggregate charges and exhibit excellent SERS properties [42,43,44,45,46,47]. In addition, semiconductor–metal bonding has also been used to achieve highly sensitive SERS substrates, owing mainly to the charge transfer between the substrates and the probe molecule through the conduction band of the semiconductor, which is a chemical enhancement (CM) [48,49]. Semiconductors have an energy gap between the full valence band (VB) and the empty conduction band (CB); therefore, the charge transfer between semiconductor nanomaterials and molecules depends on the coupling between their energy levels (including CB, VB, highest occupied molecular orbital (HOMO), and lowest unoccupied molecular orbital (LUMO)) and their energy levels. As an environmentally friendly material, zinc oxide (ZnO) is recognized as an important photocatalyst characterized by a hexagonal fibrillated zincite structure and a wide forbidden band of 3.37 eV [50,51]. The appropriate amount of ZnO loading acts as an electron transfer bridge between the silver Fermi energy level and the LUMO of the molecule, which is the key to play its important and bridging role in the CT effect. A popular approach is to load a small amount of noble metal on the semiconductor surface to improve the SERS performance of the substrates by introducing a small amount of EM in the CM, which saves costs. However, it has been shown that EM can contribute about 11 orders of magnitude to the enhancement factor of noble metals. For semiconductors, SERS enhancement is dominated by CM with values of only 10–10^3^ [48,52,53]. To fully utilize EM, this study modified large-size silver nanowires with small-size ZnO nanoparticles to prepare SERS substrates (ZnO NPs@Ag NWs), where semiconductors provide additional CM.

The univariate linear regression model (ULR) is a well-known method for quantifying aging characteristics in transformer oil [54,55,56]. This method often analyzes only one characteristic peak, ignoring the rich information contained in multiple characteristic peaks or even the full spectrum, but also the intensity of one characteristic peak used for modeling is unstable, which leads to inaccurate predictions, thus limiting its application in acetone quantification. On the other hand, stoichiometric models such as multiple linear regression (MLR) and partial least square (PLS) can be used for the analysis of multiple characteristic peaks or full spectra, which are suitable for complex multi-component spectra with high model accuracy and excellent prediction stability [57,58,59,60,61,62]. Stoichiometric methods help to extract selected relevant information about the target analyte from spectra containing fingerprints of various chemical components present in the sample [63,64]. For example, silver nanoparticles coated with zinc oxide nanoflowers (Ag@ ZnO NFs) were prepared as SERS substrates and combined with a stoichiometric method for the quantitative detection of deltamethrin in wheat, which achieved a limit of detection of 0.16 μg kg^−1^ [65]. 

This work is mainly aimed at establishing a fast and sensitive method for acetone detection in transformer oil by using the SERS technique combined with stoichiometric algorithms. The loading of ZnO nanoparticles was designed to optimize ZnO NPs@Ag NWs SERS substrates, which were used to detect acetone solution after extraction (the extractant was pure water). Thereafter, ULR, MLR, and PLS were introduced to establish quantitative regression models for acetone, and the performance of the modeling was evaluated by the root mean square error (RMSE), correlation coefficient (R), and mean absolute percentage error (MAPE). Compared with the traditional method, the method presented here is simple, accurate, and stable, which is extremely suitable for the rapid and accurate determination of trace acetone concentrations in electrical equipment of oil-paper insulation to monitor the safety and stability status of the power system in time.

## 2. Results

### 2.1. Characterization of ZnO NPs@Ag NWs

First, the elements and morphology of the Ag NWs and ZnO NPs@Ag NWs were characterized by TEM. From the TEM images (Appendix A), the pristine Ag NWs (diameter ~40 nm) intersect each other to form a network, whereby the significant enhancement of the electromagnetic field may originate from the strong localized surface plasmon resonance of the silver nanomaterial itself, as well as the coupling of the SPPs between adjacent plasmonic Ag nanowires [66]. The TEM image (Figure 1a) shows that the ZnO NPs@Ag NWs (0.02 g) still retain the morphology of pure Ag NWs, while the ZnO NPs (particle size ~22 nm) are uniformly anchored on the surface of Ag NWs. Therefore, the electromagnetic enhancement generated by the silver nanowires and the electron transfer interface of ZnO nanoparticles were formed. High-angle annular dark-field scanning transmission electron microscopy (HAADF-STEM) and energy-dispersive spectroscopy (EDS) mapping images (Figure 1b–e) demonstrate the presence of silver, zinc, and oxygen elements. It was also confirmed that the ZnO NPs were loaded on Ag NWs, where the silver elements were staggered in a linear pattern, the zinc elements were uniformly distributed near the silver elements, whilst the zinc and oxygen elements were easily stacked together. In Figure 1f, the lattice stripe spacings of Ag are 0.2314 and 0.2010 nm, corresponding to the (111) and (200) crystal planes of Ag, respectively. Meanwhile the lattice stripe spacing for ZnO is 0.2758 and 0.2413 nm, corresponding to the (100) and (101) crystal planes, respectively. As can be seen from the EDS spectra (Figure 1g), apart from the peaks of silver, zinc, and oxygen elements, and the absence of other obvious spurious peaks, the peak of assigned copper is generated by the copper network. There is a clear diffraction ring in its SAED diagram (Figure 1h), which indicates that ZnO NPs@Ag NWs have a polycrystalline structure, further confirmed by the (101) and (103) crystal faces of ZnO and the (200) and (311) crystal faces of Ag [67].

XRD was used for the physical phase analysis of Ag NWs and ZnO NPs@Ag NWs. As shown in Figure 2, with the increase of ZnCl_2_ content, the diffraction peaks of Ag did not change significantly. The diffraction peaks of Ag NWs correspond to the (111), (200), (220), (311), and (222) crystal planes, where the peaks are positioned exactly in accordance with the standard data of Ag (JCPDS 04-0783). The diffraction peaks of ZnO NPs@Ag NWs show that the intensity of ZnO diffraction peaks gradually becomes stronger as the ZnCl_2_ content increases, which indicates a gradual increase in the loading of ZnO nanoparticles. The (JCPDS 36-1451) characteristic diffraction peaks of ZnO correspond to (100), (002), (101), (100), (110), (103), (121), and (201) crystal planes [68]. These again demonstrated the presence of silver, zinc, and oxygen elements and the successful adsorption of ZnO nanoparticles onto the Ag NWs surface. As the ZnCl_2_ content decreases, the intensity of the ZnO diffraction peak becomes progressively weaker, which indicates that the loading of ZnO NPs is gradually decreasing. The absence of other elements in the test results demonstrated the high purity of the ZnO NPs@Ag NWs structure.

Similarly, XPS spectra (Figure 3a) from Zn 2p, O 1s, and Ag 3d confirm the presence of silver, zinc, and oxygen elements. XPS was also used to study the electronic states of the elements in ZnO NPs@Ag NWs. In the high-resolution XPS spectra of Ag 3d (Figure 3b), the binding energy peaks at 373.15 and 367.15 eV correspond to Ag 3d3/2 and Ag 3d5/2, respectively. The splitting energy between Ag 3d5/2 and Ag 3d3/2 is 6.0 eV, demonstrating that the Ag element in the composite is present in the Ag^0^ state. The Zn 2p spectra (Figure 3c) indicate that the element Zn is mainly present in the chemical state of Zn^2+^. The binding energy peaks of Zn 2p 1/2 and Zn 2p 3/2 are 1045.4 and 1022.3 eV, respectively, which are positively shifted by about 0.5 eV compared to 1044.9 and 1021.8 eV for pure ZnO nanoparticles. The spectrum of O 1s (Figure 3d) exhibits a clear asymmetry and can be divided into two Gaussian peaks: one at 532.65 eV for the O^2-^ assignment to the Zn-O bond, and another at 531.5 eV assigned to surface hydroxyl oxygen (OH) due to structural defects. These oxygen groups could promote the charge transfer process to increase SERS activity by effectively inhibiting the recombination of electron−hole pairs [69].

### 2.2. Optimization of ZnO NPs Loadings

The suitable ZnO NPs loading on the Ag NWs is significant to SERS performances. Firstly, we characterized the morphology of ZnO NPs@Ag NWs (0, 0.2, 0.02, and 0.005 g) with SEM (Figure 4a–d). Additionally, taking R6G as the molecular probe, the effect of the amount of ZnCl_2_ (0.2, 0.1, 0.05, 0.02, 0.01, and 0.005 g) on the SERS intensity of the composites was investigated, and the corresponding SERS spectra are shown in Figure 4e. We can see that the SERS performance of pure ZnO is far inferior to that of AgNWs, while the composite of ZnO and AgNWs significantly improves the SERS sensitivity. The CT enhancement mechanism can be explained by Appendix A, which shows: (1) the charge transfer between the Fermi energy level of the molecule and the LUMO energy level of the molecule, and (2) that the appropriate amount of ZnO loading acts as an electron transfer bridge between the Fermi energy level of the silver and the LUMO of the molecule, forming an effective “donor–bridge–acceptor” system. Therefore, the excellent surface-enhanced Raman scattering performance of silver/ZnO nanoparticles can be attributed to the synergistic CT effect of plasmonic silver, semiconductor ZnO, and molecules. As the amount of ZnCl_2_ increases, the larger the particle size of the generated ZnO particles, and the stronger the diffraction peaks of ZnO in the corresponding XRD patterns. The particle size distribution of ZnO NPs was around 7.9 nm when the amount of ZnCl_2_ was 0.005 g. Taking the peak at 1649 cm^−1^ as a reference, it can be seen that the ZnO loading effectively improved the SESR signal. ZnO NPs@Ag NWs (0.005 g) produced a 1.7-fold signal compared to pure Ag NWs (Figure 1f). When the amount of ZnCl_2_ was increased to 0.02 g, the particle size distribution of ZnO NPs was around 18.5 nm, with the best SERS performance of ZnO NPs@Ag NWs (0.02 g), which was 4.2 times higher than that of pure Ag NWs. The particle size distribution of the ZnO NPs is around 92 nm when up to 0.2 g. However, the overloading of ZnO leads to a reduction in Raman intensity, whereas the peak intensity of ZnO NPs@Ag NWs (0.2 g) is comparable to that of Ag NWs, which may be due to the silver surface hotspot being covered by too many ZnO nanoparticles, resulting in a significant reduction in the electromagnetic enhancement of silver. The optimal ZnO NPs@Ag NWs (0.02 g) would be used for the subsequent SERS detection.

The SERS performance of ZnO NPs@Ag NWs (0.02 g) was evaluated using R6G as a probe. The sensitivity of the substrates was detected with different concentrations of R6G (10^−6^ to 10^−12^ M). As shown in Figure 5a,b, the SERS intensity decreased with the decrease of R6G concentration. Interestingly, the characteristic peak at 1649 cm^−1^ was identified even at the R6G concentration of 10^−12^ M, indicating the high sensitivity of the substrates. EF is an important index to quantify the performance of the substrates. The calculated EF value is as high as 4.21 × 10^7^ using 1649 cm^−1^ as the characteristic peak, which is calculated in detail in Supporting information S2, indicating that ZnO NPs@Ag NWs are active SERS substrates with excellent enhancement performance. 

Uniformity is also an important indicator in assessing the SERS performance of the substrates. By randomly selecting 20 positions in the same substrates, the intensity of the characteristic peaks remained almost constant, as in Figure 5c, with a relative standard deviation (RSD) of only 3.32% (Figure 5d). This indicates that the substrates had good uniformity. We used Raman mapping to evaluate the homogeneity of R6G Raman spectra for a randomly selected large region (20 × 20 μm^2^). The intensity distribution of the R6G vibrational mode (1649 cm^−1^) in this region is shown in the Appendix A, and the corresponding SERS intensity histogram is shown in Appendix A. The RSD is calculated to be 9.46%. The results show that we can still obtain relatively uniform SERS signals over a large area.

Good stability is necessary for the practical application of SERS substrates. The SERS intensity of pure Ag NWs decreased to 34% of the initial intensity at 1649 cm^−1^ in the ambient environment for 30 days (Figure 6a,b). In contrast, ZnO NPs@Ag NWs substrates remained at more than 90% of the initial intensity (Figure 6c,d). This indicates that ZnO NPs@Ag NWs have superior long-term stability compared to Ag NWs, which may be due to the reduced substrates’ air oxidation by the modified ZnO nanoparticles [70,71].

### 2.3. SERS Spectral Data Analysis of Acetone

The ZnO NPs@Ag NWs substrates were further used for the practical application of acetone detection. Under the excitation of the laser, the combination of the acetone extract with the substrates produced an intense SERS signal (Figure 7a). The peak at 2930 cm^−1^ was used as the characteristic peak, which is 3.3 times more intense than the peak without substrates. Compared with ZnO NPs@Ag NWs, pure water, and pure acetone, the peaks at 800, 1074, 1238, 1360, 1425, 1699, and 2930 cm^−1^ were used as the Raman characteristic peaks of acetone in acetone extracts with ZnO NPs@Ag NWs. Then, the Raman vibrational characteristics of acetone molecules were simulated and analyzed using Gaussian software, together with the vibrational attribution of the main Raman signals. The peak signal at 800 cm^−1^ is due to C-C stretching vibration, that at 1074 cm^−1^ is due to C-C shearing motion and C-H wobble, at 1238 cm^−1^ is due to C-C and C-H shearing motion, at 1360 cm^−1^ is due to C-C stretching and C-H shearing motion, at 1425 cm^−1^ is due to C-H shearing motion, at 1699 cm^−1^ is due to C=O stretching, and that at 2930 cm^−1^ is due to C-H stretching. The SERS spectra of the acetone extracts with ZnO NPs@Ag NWs are shown in Figure 7b. The intensity of the SERS signal gradually decreases with the decrease of the acetone concentration. The inset shows the local magnifications of 0.09, 0.03, 0.01, and 0.003 mg/g, and we can also see the minimum detection concentration of 0.003 mg/g.

### 2.4. Quantitative Modeling Analysis of Acetone

Combined with the positions of the characteristic peaks of acetone extracts, Raman shifts in the ranges of 750~1800 cm^−1^ and 2700~3040 cm^−1^ were selected for analysis in this study, which are informative and clear, so that the information in the spectra can be effectively extracted. The original SERS spectra of the samples are shown in Appendix A. ULR, MLR, and PLS were used to fit the peak intensities of a single characteristic peak, multiple characteristic peaks, and the full spectrum to the acetone concentration, respectively, and the optimal quantitative regression analysis method needs to be determined.

#### 2.4.1. ULR Model

Considering the range of vibration wave number and signal sensitivity factors of the characteristic peaks of acetone and water, we chose 2930 cm^−1^ as the single characteristic peak of the linear fitting model. The characteristic peak intensity was obtained by fitting the characteristic curve using Gaussian Lorentz. Since the intensity of the acetone Raman characteristic peak was 0 at the acetone concentration, we set the intercept to 0. The quantitative analysis of acetone extracts based on ULR was performed, and the prediction results are shown in Figure 8a. The results show that R = 0.95548 and RMSE = 0.69907 mg/g, with good linear correlation, but the absolute value of the relative error between the actual and predicted concentrations was large. Especially at low concentrations, the relative errors were as high as 100%~340%. The model was not ideal, mainly because our linear regression model used only one feature peak, and the intensity of the single characteristic peak was not stable, resulting in inaccurate model predictions.

#### 2.4.2. MLR Model

To improve the accuracy of the quantitative acetone analysis model, we chose the 800 and 2930 cm^−1^ locations with high signal sensitivity as the characteristic peaks of the multiple linear regression model. We selected the acetone concentration as the dependent variable Y and the intensity of the corresponding Raman characteristic peaks at 800 and 2930 cm^−1^ as the independent variables for regression analysis to predict the acetone concentration of the samples. The predicted results are shown in Figure 8b. The results show that R is 0.98432 and the RMSE is 0.41038 mg/g, the linear correlation has improved, and the absolute value of the relative error has decreased. Again, the absolute value of the relative error at low concentrations is still large, which is due to the fact that the low concentration is close to the detection limit of Raman spectroscopy. Overall, the model effect was improved because we used two characteristic peaks for fitting, which increased the stability of the model prediction.

#### 2.4.3. PLS Model

For further analysis, we used the PLS model to establish the statistical relationship between acetone content and full-spectrum intensity, and 32 samples from the standard series of acetone extracts were used as the training set and 16 samples as the prediction set. The corresponding spectral and concentration matrices were imported into the Unscrambler as input variables, and regression analysis was performed by the PLS method. The PLS model built for acetone spectra generated R_c_ = 0.992273 and RMSECV = 0.214842 mg/g values in the calibration set, and R_p_ = 0.980813 and RMSEP = 0.327962 mg/g values in the prediction set.

Suitable preprocessing methods can improve the accuracy of the model. Different preprocessing methods were tried in this model, as shown in Appendix A. The performances of both training and prediction sets were basically improved after using preprocessing algorithms such as SG, SNV, and Baseline. This is because Baseline can well-eliminate baseline translation and drift, SG processing can effectively eliminate the interference of background noise, and SNV can eliminate the effect of light range variation and surface scattering phenomenon on the spectrum. The above discussion reveals that SG, SNV, and Baseline pretreatment methods can retain the absolute SERS peak intensities and are ultimately used to develop the best model for predicting acetone extract concentrations. The optimum principal component numbers (PCs) were selected by cross-validation ten times according to the minimum value of RMSECV. The RMSECV was minimized when the number of primary factors was 5 (Figure 8c). Therefore, the experiment selected the optimal principal component fraction of 5. The PLS model built for acetone spectra generated R_c_ = 0.99729 and RMSECV = 0.12596 mg/g values in the calibration set, and R_p_ = 0.997678 and RMSEP = 0.11408 mg/g values in the prediction set, as displayed in Figure 8d. Compared with ULR and MLR models, the goodness of fit was improved, the root mean square error decreased substantially, and the relative error of prediction for standard solutions with smaller concentrations was substantially reduced. In particular, for the standard solutions of 7.29 mg/g, the absolute values of the prediction errors were all around 1.5%. 

#### 2.4.4. Comparison of Quantitative Models

A comprehensive analysis of R, RMSE, and MAPE at 7.29 and 0.003 mg/g for the three quantitative analysis models was performed to obtain the best model for rapid and accurate quantification of acetone, and the results are shown in Table 1. It is observed that all three quantitative models were satisfactorily fitted. However, the linear fitting model was the worst predictor for the high concentration of acetone solution, while the multiple linear regression model was the worst predictor for the low concentration. The least squares regression model afforded the most stable prediction for the standard series of acetone solutions, with a limit of quantification value of 0.003 mg/g. In addition, the limit of detection was calculated to be 1 μg/g by considering the signal-to-noise ratio (3S/N) criterion. This method obtained higher correlation coefficients and lower root mean square errors, as well as the best prediction for both high and low concentrations. Overall, the partial least squares regression model was chosen as the best for the quantitative analysis of acetone in SERS.

## 3. Materials and Methods

### 3.1. Materials

Silver nanowires suspension (Ag NWs, CST-NW-S30) in ethyl alcohol was purchased from Suzhou Coldstones Technology Co., Ltd., Suzhou, China. Rhodamine 6 g (R6G, ≥95%), zinc chloride (ZnCl_2_), N,N-dimethylformamide (DMF), and sodium hydroxide (NaOH) were obtained from Shanghai Aladdin Reagent Co., Ltd, Shanghai, China. Anhydrous alcohol (99%) and Millipore ultrapure water were purchased from Sinopharm Chemical Reagent Co., Ltd, Shanghai, China. Pure acetone and Kelian 25# transformer mineral oil (41.6% naphthenic, 50.0% paraffinic, and 8.4% aromatic) were purchased from the Chuanrun Lubricant Company. All reagents were of analytical grade and used without further purification. Water used in all experiments was purified on a Millipore system (18.2 MΩ cm).

### 3.2. Apparatus

High-resolution transmission electron microscopy (HRTEM), selected area electron diffraction (SAED), morphology, and elemental characterization of the products were performed on the transmission electron microscope (TEM, Talos F200S, Prague, Czech Republic). The bulk phase structure of the material was characterized in the 2θ range from 20° to 90° by powder X-ray diffraction (XRD, PANalytical X’Pert Powder, Amsterdam, Netherlands). The chemical valence states of the elements on the substrates’ surface were analyzed by X-ray photoelectron spectroscopy (XPS, ESCALAB 250Xi, London, UK). The microscopic characteristics of the materials were analyzed using environmental scanning electron microscopy (SEM, Quattro S, New York, NY, US). In this work, all SERS experiments were conducted using a confocal laser Raman spectroscopy detection platform that was built by our group. Ag substrates are recommended to be tested using a 532 nm Raman system, so a 532 nm solid-state laser was chosen as the light source with a maximum laser power of 100 mW, of which 70 and 50 mW were chosen for the detection of molecular probes R6G and acetone, respectively. A CCD (refrigeration temperature −85 °C, resolution 2000 × 256, quantum efficiency > 90%) was attached to the spectrometer, and a 50× objective and a grating (1200 lines/750 nm) were used in the spectrometer. The integration time of the spectrometer was set to 0.1 s, and the cumulative integration was 100 times.

### 3.3. Synthesis of ZnO NPs@Ag NWs Composite Materials

ZnO NPs@Ag NWs composite materials were prepared according to the previous method with slight modifications [70]. ZnCl_2_ (0.2, 0.1, 0.05, 0.02, 0.01, and 0.005 g) was dissolved in DMF and then mixed with 50 mg of Ag NWs and stirred well. The pH of the solution was adjusted to neutral by using NaOH, after which the solution was heated to 100 °C with stirring for 0.5 h. The final dark gray color products were collected by centrifugation and repeatedly washed with water and ethanol.

### 3.4. SERS Detection

R6G was used as a molecular probe to evaluate the basic SERS performance. The ZnO NPs@Ag NWs were used as the SERS substrates, which were adsorbed with R6G ethanol solution and then transferred to clean silicon wafers for SERS detection. SESR performance of ZnO NPs@Ag NWs composites with different ZnO loadings was evaluated to obtain the optimal ZnO NPs@Ag NWs composites. The optimal ZnO NPs@Ag NWs composites were used as substrates for subsequent SERS detection. To assess substrates’ SERS activity and sensitivity, the enhancement factor (EF) was calculated. To assess the uniformity of the substrates, 20 locations of the substrates were randomly selected for SESE detection. Chemical stability was studied on substrates maintained on silicon wafers at room temperature for one month, and SERS spectra were measured every other day under the same conditions, compared with the chemical stability of pure Ag NWs.

To detect acetone, standard solutions of acetone in transformer oil (7.29, 2.43, 0.81, 0.27, 0.09, 0.03, 0.01, and 0.003 mg/g) were extracted with pure water. The ZnO NPs@Ag NWs materials were dispersed in the alcohol solution and spin-coated onto silicon wafers. After natural evaporation, they were used as SERS substrates. The prepared silica-based ZnO NPs@Ag NWs substrates were placed in the extracted acetone-transformer oil solution, and they were transferred into closed quartz cuvettes with a 3 mm optical path for SERS detection under the same conditions. For the standard solutions of acetone extract, six samples were prepared. For each sample, six spectrums were collected at different positions. 

### 3.5. Stoichiometric Models

Various stoichiometric methods have been applied to assess the spectral characteristics of specific compounds and to quantify the spectra. This study aimed to quantify acetone in transformer oil using multiple characteristic quantity analysis methods (e.g., MRL) and full spectral analysis methods (e.g., PLS), in comparison to the commonly used spectral quantification methods (MLR).

#### 3.5.1. MRL Models

The multiple linear regression method is the most conventional quantitative analysis method in multiple linear regression, which is mainly used for revealing the statistical relationship between the single dependent variable and multiple independent variables. The expression of the regression model is:Y = β_0_ + β_1_x_1_ + β_2_x_2_ + ⋯β_p_x_p_ + ε(1)
where β_0_ is the constant term or intercept, β_p_ is the partial regression coefficient, and ε is the random error.

In the MRL model, the selection of appropriate independent variables is one of the prerequisites for accurate prediction. Its model evaluation parameters are expressed in terms of R, and RMSE and MAPE are used to express the predictive stability of the model. The calculation formulas for RMSE and MAPE are as follows:(2)$$\mathrm{RMSE}=\sqrt{\frac{\sum_{i=1}^{n}{\left(y_{i}^{\wedge }-y_{i}\right)}^{2}}{n}}$$
where *n* is the number of samples in the model, $y_{i}$ is the standard concentration of sample *i* in the standard series, and $y_{i}^{\wedge }$ is the estimated concentration of the *i*th sample in the standard series.
(3)$$\mathrm{MAPE}=\frac{100\%}{m}\sum_{i=1}^{m}\left|\frac{y_{i}^{\wedge }-y_{i}}{y_{i}}\right|$$
where *m* is the number of certain samples in the model, $y_{i}$ is the standard concentration of sample *i* in the standard series, and $y_{i}^{\wedge }$ is the estimated concentration of the *i*th sample in the standard series. 

When the RMSE and MAPE are smaller than 1 and R is 1, the analysis of the model is more effective.

#### 3.5.2. PLS Models

The partial least squares method, a multivariate statistical analysis method with dimensions’ reduction, importance analysis of each observation, and discriminant analysis, extracts the main factors from the attributes to be measured and then ranks them by correlation to derive the optimal model. Prior to modeling, a calibration set and a prediction set are required. Then, 48 datasets of the standard solutions were randomly divided into two subsets in a 2:1 ratio. In this manner, the calibration set contains 32 datasets for model training, while the prediction set consists of 16 datasets. Its model evaluation parameters are expressed in terms of corrected correlation coefficients (R_c_) and prediction correlation coefficients (R_p_), and the predictive power of the model is expressed in terms of root mean square error of correction (RMSECV), root mean square error of prediction (RMSEP), and MAPE. The calculation equations for RMSECV and RMSEP are as follows:(4)$$\mathrm{RMSECV}=\sqrt{\frac{\sum_{i=1}^{c}{\left(y_{i}^{\wedge }-y_{i}\right)}^{2}}{c}}$$
where *c* is the number of samples in the calibration set, $y_{i}$ is the standard concentration of sample *i* in the standard series, and $y_{i}^{\wedge }$ is the estimated concentration of the sample *i* in the standard series. The model is built with the sample *i* excluded.
(5)$$\mathrm{RMSEP}=\sqrt{\frac{\sum_{i=1}^{p}{\left(y_{i}^{\wedge }-y_{i}\right)}^{2}}{p}}$$
where *p* is the number of samples in the prediction set, $y_{i}$ is the standard concentration of sample *i* in the prediction set, and $y_{i}^{\wedge }$ is the estimated concentration of the model for the sample *i* in the prediction set.

When the RMSECV, RMSEP, and MAPE are smaller and the R_c_ and R_p_ converge to 1, the analysis of the model is better.

Raw SERS spectra contain chemically relevant information as well as signals from systematic interferences. Interference from environmental and equipment noise is difficult to avoid, especially interference such as background fluorescence, which leads to baseline drift, severely degrades the quality of Raman spectra, and decreases the accuracy of substance detection [61,72,73]. Therefore, the accuracy of the model can also be further improved by optimizing the correction and prediction sets through spectral preprocessing methods such as equal Savitzky–Golay smoothing filtering (SG), standard normal variate transformer (SNV), baseline removal (Baseline), multiplicative scatter correction (MSC), and first difference (D1).

## 4. Conclusions

In this study, a simple and accurate method for the quantitative detection of acetone in oil-paper insulating materials by SERS coupled with PLS was developed. ZnO nanoparticles were successfully anchored on Ag NWs to obtain ZnO NPs@Ag NWs substrates with excellent SERS performance, in addition to the ZnO NPs modification remarkably improving the SERS stability of the silver nanowires. The SERS intensity of ZnO NPs@Ag NWs remained above 90% of the initial intensity in the ambient environment for 30 days, compared to 34% for Ag NWs. Using rhodamine 6G (R6G) as the probe molecule, the limit of detection was as low as 10–12 M and the enhancement factor (EF) was as high as 4.21 × 107 based on the optimal (ZnO NPs@Ag NWs). In addition, the relative standard deviation (RSD) was only 3.32% for the intensity of the characteristic peaks at ten different positions.

ZnO NPs@Ag NWs were successfully applied as SERS substrates for the detection of acetone in transformer oil. The SERS spectra were quantified using three regression algorithms of ULR, MRL, and PLS. The experimental results showed that the SERS combined with the partial least squares model obtained more accurate predictions of acetone concentrations, and the model accuracy was further improved using SG, SNV, Baseline, and other preprocessing methods, which produced R_c_ = 0.99729, RMSECV = 0.12596 mg/g, R_p_ = 0.997678, RMSEP = 0.11408 mg/g, MAPE (7.29 mg/g) = 1.6%, and MAPE (0.003 mg/g) = 18.47%. The limit of detection was 1 μg/g, and the limit of quantification was 0.003 mg/g, which easily met the requirements of engineering testing. The model was highly accurate, with better prediction, by which the established method was competent for the quantitative analysis of acetone in transformer oil. Based on this study, the number of samples can be expanded, more pretreatment methods can be selected, and new modeling algorithms can be introduced to improve the accuracy of the model. We believe that this study will advance the application of SERS to rapid acetone detection in power sites and provide a strong guarantee for power equipment maintenance.

## Figures and Tables

**Figure 1 ijms-23-13633-f001:**
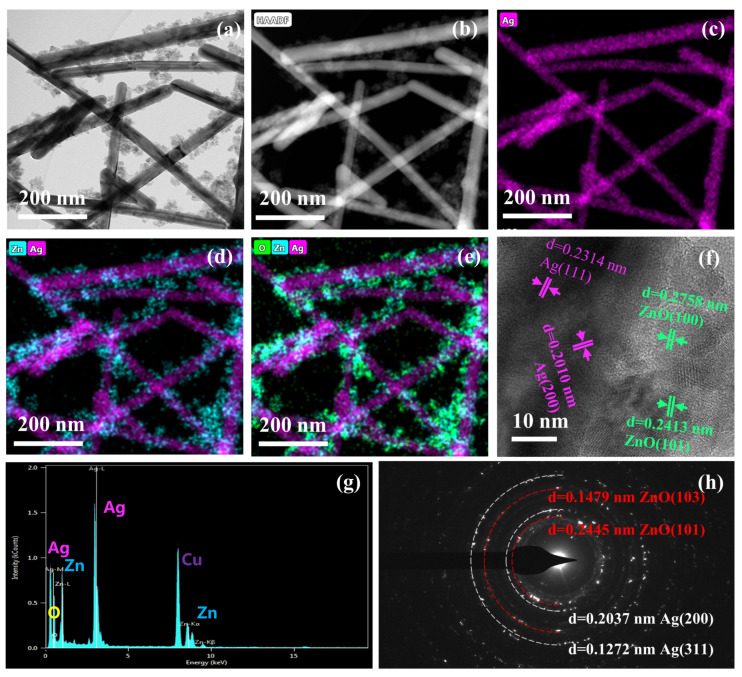
Characterization of ZnO NPs@Ag NWs (0.02 g) by transmission electron microscopy. TEM (**a**), HAADF-STEM (**b**), EDS mapping images (**c**–**e**), HRTEM (**f**), EDS spectrum (**g**), and SAED pattern (**h**) of ZnO NPs@Ag NWs (0.02 g).

**Figure 2 ijms-23-13633-f002:**
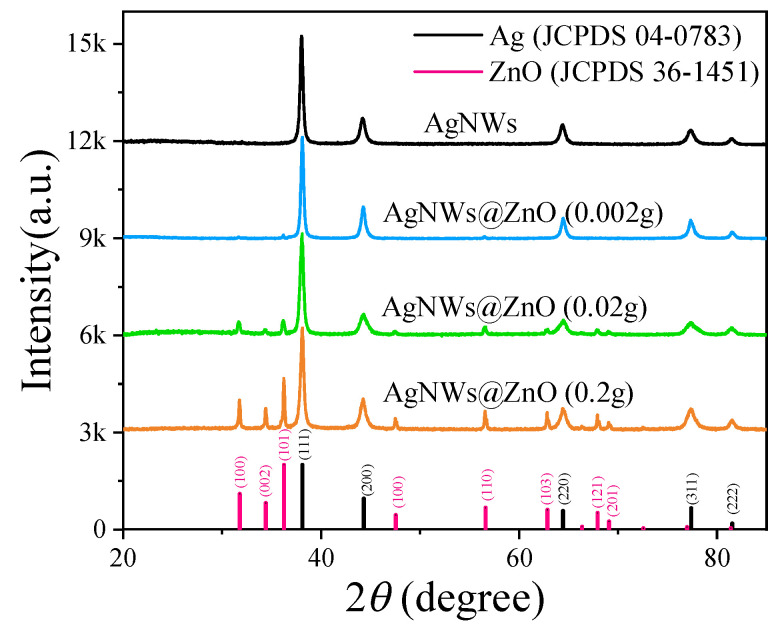
XRD patterns of Ag NWs and ZnO NPs@Ag NWs.

**Figure 3 ijms-23-13633-f003:**
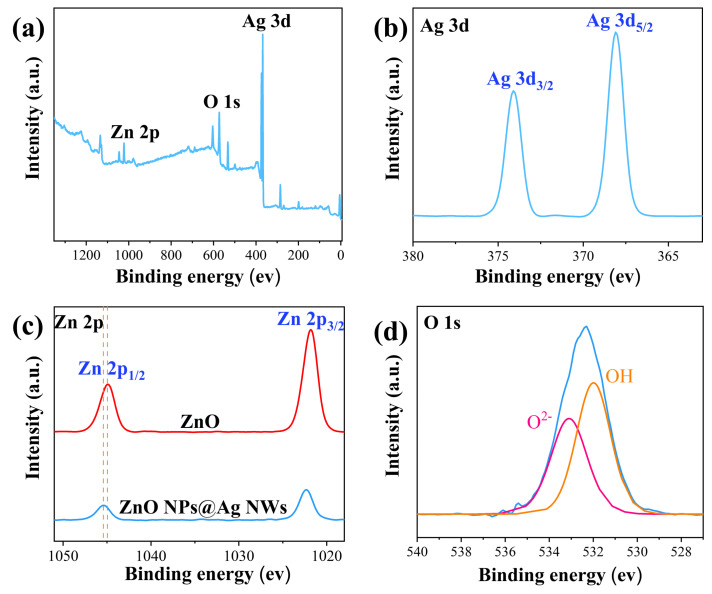
(**a**) XPS of ZnO NPs@Ag NWs. (**b**–**d**) High-resolution XPS spectra of Ag 3d, Zn 2p, and O 1s.

**Figure 4 ijms-23-13633-f004:**
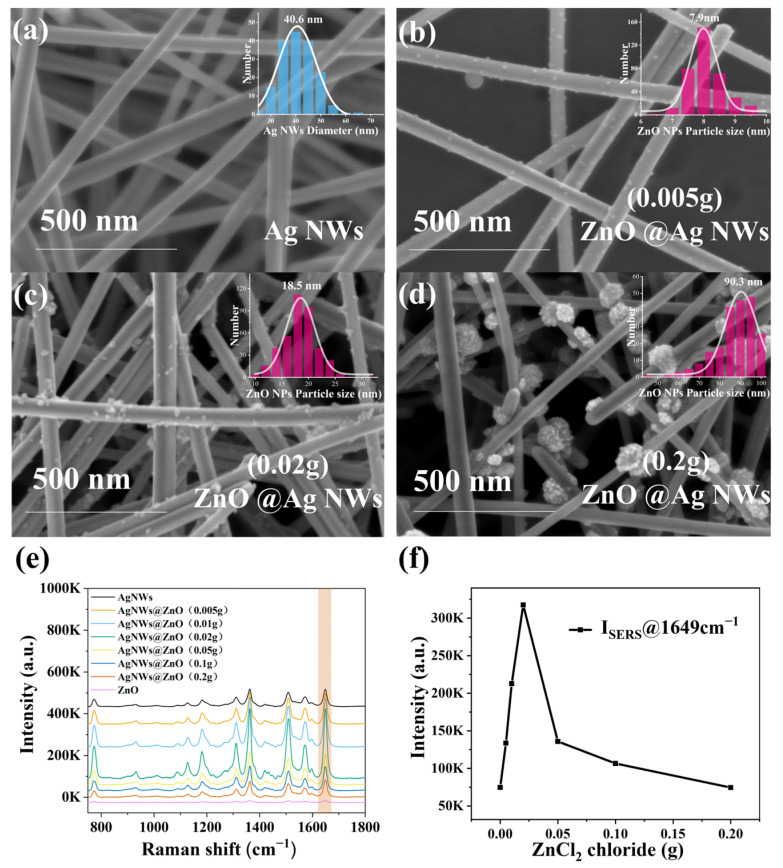
(**a**) SEM images of Ag NWs, and the inset shows the diameter distribution of Ag NWs. (**b**–**d**) SEM images of ZnO NPs@Ag NWs (0.005, 0.02, and 0.2 g), and the insets are the particle size distribution of the loaded ZnO NPs. (**e**) SERS spectra of 10^−6^ M R6G adsorbed on Ag NWs, ZnO NPs@Ag NWs, and ZnO. (**f**) Dependence of the peak intensity value at 1649 cm^−1^ with various ZnCl_2_.

**Figure 5 ijms-23-13633-f005:**
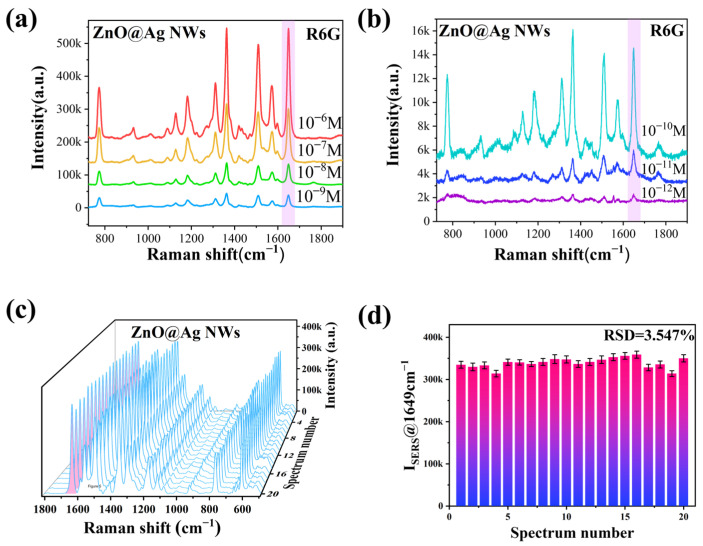
SERS spectra of R6G at a concentration of 10^−6^ to 10^−9^ M (**a**) and 10^−10^ to 10^−12^ M (**b**). SERS spectra (**c**) and uniformity (**d**) of 10^−6^ M R6G on ZnO NPs@Ag NWs at 20 random positions.

**Figure 6 ijms-23-13633-f006:**
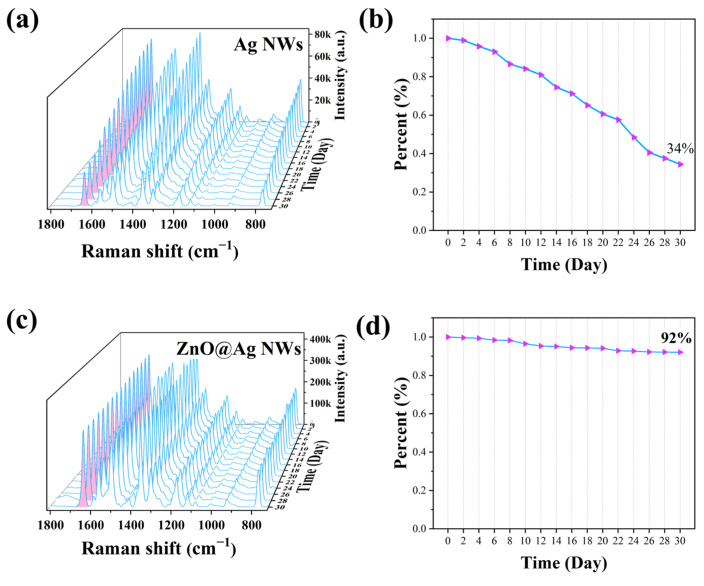
SERS spectra: long-term stability of 10^−6^ M R6G on Ag NWs (**a**,**b**) and ZnO NPs@Ag NWs (**c**,**d**) was measured every two days during one month at room temperature.

**Figure 7 ijms-23-13633-f007:**
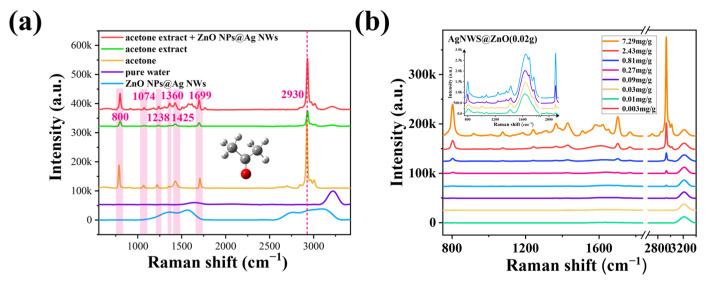
(**a**) Raman spectra of ZnO NPs@Ag NWs substrates, pure water, acetone, acetone extract (extractant was pure water), and acetone extract enhanced by ZnO NPs@Ag NWs. (**b**) SERS spectra of different acetone extract concentrations (the inset shows the local magnifications of 0.09, 0.03, 0.01, and 0.003 mg/g).

**Figure 8 ijms-23-13633-f008:**
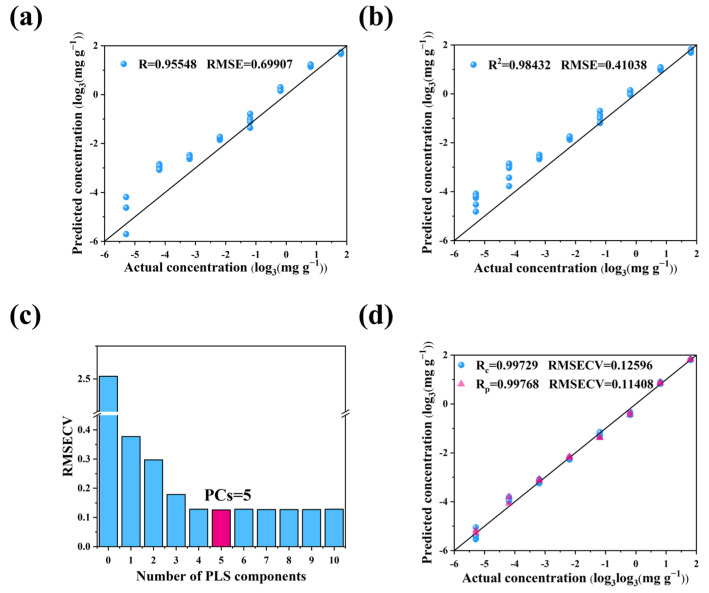
Analytical results of models. Scatter plots of predicted concentration (log_3_ (mg·g^−1^)) vs. actual concentration (log_3_ (mg·g^−1^)) for ULR (**a**), MLR (**b**), and PLS (**d**) models. (**c**) RMSECV vs. PLS components for the PLS model.

**Table 1 ijms-23-13633-t001:** Comparison of three quantitative models.

Model		R	RMSE	MAPE	
				7.29 mg/g	0.003 mg/g
ULR		0.95548	0.69907	11.47%	107.03%
MLR		0.98432	0.41038	6.12%	199.61%
PLS	calibration	0.99729	0.12596	1.33%	30.13%
	prediction	0.99768	0.11408	1.60%	18.47%

## Data Availability

The data presented in this paper are available upon request from the corresponding author.

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
