# Peer review of "Quantitative Analysis of Acetone in Transformer Oil Based on ZnO NPs@Ag NWs SERS Substrates Combined with a Stoichiometric Model"

_ijms, 2022, doi:10.3390/ijms232113633_

Round 1
Reviewer 1 Report
The work is of both scientific and practical interest. Numerical estimates and the result obtained are well-founded, in particular, TEM images and measurement results are presented. I consider it expedient to publish the article.
Author Response
Reply to review comments
Manuscript ID: ijms-1927589 Type: article
Title: Quantitative analysis of acetone in transformer oil based on ZnO NPs@Ag NWs SERS substrates combined with stoichiometric model
Author: Xinyuan Zhang; Chongqing University
Dear Reviewers,
Thank you very much for giving us this opportunity to revise our manuscript “Quantitative analysis of acetone in transformer oil based on ZnO NPs@Ag NWs SERS substrates combined with stoichiometric model” (No.: ijms-1927589). We have revised our manuscript according to your comments point-by-point. All the revised parts were marked in yellow highlight in the Main Text. We hope our revised manuscript can meet the requirements for International Journal of Molecular Sciences. Thank you very much for your consideration and your help.
Reviewer #1:
Comments to the Author
The work is of both scientific and practical interest. Numerical estimates and the result obtained are well-founded, in particular, TEM images and measurement results are presented. I consider it expedient to publish the article.
Response 1: Thank you for your encouraging comment!

Reviewer 2 Report
In this paper, the authors report on qualitative analysis of low acetone concentration using a nanohybrid system based on ZnO nanoparticles and Ah nanowires. The paper is in general, well structured. Questions and recommendations are mentioned below:
1. The introduction part didn't provide a clear statement on the advantages of choosing ZnO nanoparticles as the SERS template as well in particularly about AgNWs. The authors should provide a clear benefit of the SERS substrate structure in this work compared to previous literature.
https://doi.org/10.1016/j.jphotochem.2022.114123
https://doi.org/10.1016/j.apsusc.2017.01.216
2. concerning Fig 1. How did the authors justify the degree of compactness of Ag nanostructures? It is a random orientation with a cross-link at certain places. The places at cross-link will get good EM enhancement rather than the individual AgNWs. How could the randomly distributed AgNWs maintain uniformity in signal sensitivity? please clarify it.
3. Looking for XRD data on whether there is any shift in the Ag peaks upon ZnO loadings. Please comment and explain. Please report the position and the width of the most important lines, including their standard deviation. Please discuss if these parameters depend on concentration. Also, comment on the correlation between particle size of ZnO variation (Fig 4 a) with XRD data.
4. I think the author should at least mention at which wavelength the SERS tests are conducted? Substantiate for selection of particular lasing wavelength.
5. On comparing the sensing tests associated with the prepared SERS substrates, the authors checked on the performance of pure ZnO nanoparticles alone (Zn percent 0.02g) for R6G detection.
6. In SERS experiments, what is the error associated when evaluating structures processed in the same batch but in different spatial locations of the substrate? What are the errors associated to the sensing performances from batch to batch?
7. As the experiments reported were not designed as absolute measurements, the author cannot discuss intensities or areas. However, the author may represent and discuss relative intensities or/and areas (Fig. 5 & 6). This is one of the main difficulties in assessing SERS. A demonstration is required.
8. Did you also notice any type of thermalization behavior when performing measurements for stability purpose analysis?
9. References: require updated
10. The authors need to pay attention to typo errors. Please check the manuscript, as a lot of mistakes and errors are found. Please try to improve your overall English as well. For example, in line 278, please correct ‘SESR signal’ as ‘SERS signal’. In line 278, correct ‘znO NPs@Ag NWs (0.005g)’ as ‘ZnO NPs@Ag NWs (0.005g)’. In line 279, correct ‘zncl2’ as ‘ZnCl2’. In line 296, correct ‘R6g concentration’ as ‘R6G concentration’
11. In Figure 3d, XPS spectrum for oxygen is marked wrongly as it is mentioned in the discussion part ‘The spectrum of O 1s exhibits a clear asymmetry and can be divided into two Gaussian peaks: one at 531.5 261 eV for the O2- assignment to the Zn-O bond, and another at 532.65 eV assigned to surface 262 hydroxyl oxygen (OH) due to structural defects. Please correct the position of O2- and OH in the spectra
12. For the XPS spectra of Ag (Figure 3b), no description was given in the discussion. Please explain the Ag peaks and significance
13. What about the UV-visible absorbance of the fabricated substrates? Please include absorbance or reflectivity data
14. The enhanced SERS activities of ZnO NPs@Ag NWs should be explained in terms of the charge transfer mechanism. The functional properties of ZnO should be explained in the introduction section.
15. Sample ID given in Figure 4e should be modified, especially the font size, so it will be readable. Also, modify Figure 2 (make the indexing to be readable)
Author Response
Reply to review comments
Manuscript ID: ijms-1927589 Type: article
Title: Quantitative analysis of acetone in transformer oil based on ZnO NPs@Ag NWs SERS substrates combined with stoichiometric model
Author: Xinyuan Zhang; Chongqing University
Dear Reviewers,
Thank you very much for giving us this opportunity to revise our manuscript “Quantitative analysis of acetone in transformer oil based on ZnO NPs@Ag NWs SERS substrates combined with stoichiometric model” (No.: ijms-1927589). We have revised our manuscript according to your comments point-by-point. All the revised parts were marked in yellow highlight in the Main Text. We hope our revised manuscript can meet the requirements for International Journal of Molecular Sciences. Thank you very much for your consideration and your help.
Reviewer #2:
Comments to the Author
In this paper, the authors report on qualitative analysis of low acetone concentration using a nanohybrid system based on ZnO nanoparticles and Ah nanowires. The paper is in general, well structured. Questions and recommendations are mentioned below:
1. The introduction part didn't provide a clear statement on the advantages of choosing ZnO nanoparticles as the SERS template as well in particularly about AgNWs. The authors should provide a clear benefit of the SERS substrate structure in this work compared to previous literature.
https://doi.org/10.1016/j.jphotochem.2022.114123;https://doi.org/10.1016/j.apsusc.2017.01.216
Response: We are very grateful for your suggestions. According to the suggestions of reviewer, we have modified the corresponding positions in the paper. We have added the advantages of ZnO and AgNWs as SERS substrates in the introduction section and cited related articles.
(A filling scheme has been added to the revised manuscript, lines 61-64, 68-77 on the page 2.)
Silver nanowires (Ag NWs) are unique plasmonic nanomaterials that support propa-gating surface plasmon excitations (SPPs) because the sharp edges or branches of silver metal nanostructures can aggregate charges and exhibit excellent SERS properties[42–47].
Semiconductors have an energy gap between the full valence band (VB) and the empty conduction band (CB), therefore, the charge transfer between semiconductor nano-materials and molecules depends on the coupling between their energy levels (includ-ing CB, VB, highest occupied molecular orbital (HOMO) and lowest unoccupied molecular orbital (LUMO)) and their energy levels. As an environmental friend endly material, zinc oxide (ZnO) is recognized as an important photocatalyst characterized by a hexagonal fibrillated zincite structure and a wide forbidden band of 3.37 eV[50,51]. The appropriate amount of ZnO loading acts as an electron transfer bridge between the silver Fermi energy level and the LUMO of the molecule, which is the key to play its important and bridging role in the CT effect. The appropriate amount of ZnO loading acts as an electron transfer bridge between the silver Fermi energy level and the LUMO of the molecule, which is the key to play its important and bridging role in the CT effect.
2. concerning Fig 1. How did the authors justify the degree of compactness of Ag nanostructures? It is a random orientation with a cross-link at certain places. The places at cross-link will get good EM enhancement rather than the individual AgNWs. How could the randomly distributed AgNWs maintain uniformity in signal sensitivity? please clarify it.
Response: Thank you for the comment. As you said, the distribution of AgNWs is random, so we double-checked the homogeneity of the SERS signal sensitivity by Raman mapping, and have included the results of Raman mapping in the revised manuscript. In our work, we use Raman mapping to evaluate the homogeneity of R6G Raman spectra for a randomly selected large region (20 × 20 μm2). The intensity distribution of the R6G vibrational mode (1649 cm-1) in this region is shown in the Supporting Information (Figure S3), and the corresponding SERS intensity histogram is shown in Figure S13B. The RSD is calculated to be 9.46%. The results show that even with the random distribution of AgNW in AgNW@ZnO, we can still obtain a relatively uniform SERS signal over a large range.
It is also worth noting that the RSD from separated spectra can be better since the focal spot can be achieved on every point/position. On the contrary, only one overall “best” focus will be used during the Raman mapping when the Raman data is collected on a much larger area (hundreds of μm2). Therefore, the RSD value calculated from the Raman mapping could be a little larger than those from individual Raman spectra.
(A filling scheme has been added to the revised manuscript, lines 368 - 376 on the page 9, we place Fig. S3. In Supporting information)
We use Raman mapping to evaluate the homogeneity of R6G Raman spectra for a randomly selected large region (20 × 20 μm2). The intensity distribution of the R6G vibrational mode (1649 cm-1) in this region is shown in the Supporting Information (Figure S3), and the corresponding SERS intensity histogram is shown in Figure S13B. The RSD is calculated to be 9.46%. The results show that we can still obtain relatively uniform SERS signals over a large area.
Figure S3. (a) SERS map of R6G at 1649 cm−1; and corresponding histograms (b) for the above mapping area.
Raman mapping spectra was obtained by using a Renishaw in Via Raman microscope with a 600 lines/mm grating and a 514 nm laser. The incident laser beam was focused by a 50 ×objective and the laser power on the samples was kept 0.5 mW to avoid laser induced heating.
3. Looking for XRD data on whether there is any shift in the Ag peaks upon ZnO loadings. Please comment and explain. Please report the position and the width of the most important lines, including their standard deviation. Please discuss if these parameters depend on concentration. Also, comment on the correlation between particle size of ZnO variation (Fig 4 a) with XRD data.
Response: Thank you for the comment. According to the suggestions of reviewer, we have modified the corresponding positions in the paper.
(A filling scheme has been added to the revised manuscript, lines 266-281 on the page 19.)
XRD was used for the physical phase analysis of Ag NWs and ZnO NPs@Ag NWs. As shown in Figure 2, With the increase of ZnCl2 content, the diffraction peaks of Ag did not change significantly. The diffraction peaks of Ag NWs correspond to the (111), (200), (220), (311), and (222) crystal planes, where the peaks are positioned exactly in accordance with the standard data of Ag (JCPDS 04-0783). The diffraction peaks of ZnO NPs@Ag NWs show that the intensity of ZnO diffraction peaks gradually be-comes stronger as the ZnCl2 content increases, which indicates a gradual increase in the loading of zno nanoparticles.The (JCPDS 36-1451) characteristic diffraction peaks of ZnO, corresponding to (100), (002), (101), (100), (110), (103 ), (121), (201) crystal planes.[72]. These again demonstrated the presence of silver, zinc, and oxygen ele-ments and the successful adsorption of ZnO nanoparticles onto the Ag NWs surface. As the ZnCl2 content decreases, the intensity of the ZnO diffraction peak becomes pro-gressively weaker, which indicates that the loading of ZnO NPs is gradually decreasing. The absence of other elements in the test results demonstrated the high purity of the ZnO NPs@Ag NWs structure.
4. I think the author should at least mention at which wavelength the SERS tests are conducted? Substantiate for selection of particular lasing wavelength.
Response: Thank you for the comment. The SERS tests were performed at 532 nm laser wavelength. According to reference[1], Ag substrates are recommended to be tested using 532nm Raman system, while 785nm is suitable for testing Au substrates. Therefore, the 532 nm Raman system used in this project was considered comprehensively for SERS testing. And it is added in the corresponding position in the text
5. On comparing the sensing tests associated with the prepared SERS substrates, the authors checked on the performance of pure ZnO nanoparticles alone (Zn percent 0.02g) for R6G detection.
Response: We are very grateful for your suggestions. According to the suggestions of reviewer, we have modified the corresponding positions in the paper. In Figure 4e, we checked on the performance of pure ZnO nanoparticles alone (Zn percent 0.02g) for R6G detection.
(A filling scheme has been added to the revised manuscript, lines 314-318 on the page 7.)
Also, taking R6G as the molecular probe, the effect of the amount of ZnCl2 (0.2 g, 0.1 g, 0.05 g, 0.02 g, 0.01 g, 0.005 g) on the SERS intensity of the composites was investigated, the corresponding SERS spectra are shown in Figure 4e. We can see that the SERS per-formance of pure ZnO is far inferior to that of AgNWs, while the composite of ZnO and AgNWs significantly improves the SERS sensitivity.
6. In SERS experiments, what is the error associated when evaluating structures processed in the same batch but in different spatial locations of the substrate? What are the errors associated to the sensing performances from batch to batch?
Response: Thank you for the comment. In the SERS experiments, the relative error of the same batch is due to the distribution of AgNws structure, the size and distribution of ZnO particles, and the thickness of the material coating at different locations. The relative errors of different batches are due to the air pressure, temperature, humidity and the thickness of the material applied in different batches during the synthesis process.
7. As the experiments reported were not designed as absolute measurements, the author cannot discuss intensities or areas. However, the author may represent and discuss relative intensities or/and areas (Fig. 5 & 6). This is one of the main difficulties in assessing SERS. A demonstration is required.
Response: We are very grateful for your suggestions. According to the suggestions of reviewer, we have modified the corresponding positions in the paper. We use Raman mapping to evaluate the homogeneity of R6G Raman spectra for a randomly selected large region (20 × 20 μm2). The intensity distribution of the R6G vibrational mode (1649 cm-1) in this region is shown in the Supporting Information (Figure S3), and the corresponding SERS intensity histogram is shown in Figure S13B. The RSD is calculated to be 9.46%. The results show that we can still obtain relatively uniform SERS signals over a large area.
(A filling scheme has been added to the revised manuscript, we place Fig. S3. In Supporting information.)
Figure S3. (a) SERS map of R6G at 1649 cm−1; and corresponding histograms (b) for the above mapping area.
Raman mapping spectra was obtained by using a Renishaw in Via Raman microscope with a 600 lines/mm grating and a 514 nm laser. The incident laser beam was focused by a 50 ×objective and the laser power on the samples was kept 0.5 mW to avoid laser induced heating.
8. Did you also notice any type of thermalization behavior when performing measurements for stability purpose analysis?
Response: Thank you for the comment. Since we were conducting Raman tests for a short time of 0.5 seconds, the laser did not cause significant thermal behavior, so this was negligible.
9. References: require updated
Response: We are very grateful for your suggestions. According to the suggestions of reviewer, we have modified the corresponding positions in the paper. We have updated the relevant references.
(A filling scheme has been added to the revised manuscript.)
- Columbus, S.; Hammouche, J.; Ramachandran, K.; Daoudi, K.; Gaidi, M. Assessing the Efficiency of Photocata-lytic Removal of Alizarin Red Using Copper Doped Zinc Oxide Nanostructures by Combining SERS Optical Detection. Journal of Photochemistry and Photobiology A: Chemistry 2022, 432, 114123, doi:10.1016/j.jphotochem.2022.114123.
- Ramachandran, K.; Muthukumarasamy, A.; Baskaran, B.; Chidambaram, S. Optical and Electrical Characteris-tics of N-ZnSmO/p-Si Heterojunction Diodes. Applied Surface Science 2017, 418, 312–317, doi:10.1016/j.apsusc.2017.01.216.
- Zhang Ren. Spectral analysis of several Raman spectroscopy-bearing substrates for biological samples and substrate selection criteria[J]. Shanghai Metrology and Testing,2018,45(2):9-1215
10. The authors need to pay attention to typo errors. Please check the manuscript, as a lot of mistakes and errors are found. Please try to improve your overall English as well. For example, in line 278, please correct ‘SESR signal’ as ‘SERS signal’. In line 278, correct ‘znO NPs@Ag NWs (0.005g)’ as ‘ZnO NPs@Ag NWs (0.005g)’. In line 279, correct ‘zncl2’ as ‘ZnCl2’. In line 296, correct ‘R6g concentration’ as ‘R6G concentration’
Response: We are very grateful for your suggestions. According to the suggestions of reviewer, we have modified the corresponding positions in the paper.
11. In Figure 3d, XPS spectrum for oxygen is marked wrongly as it is mentioned in the discussion part ‘The spectrum of O 1s exhibits a clear asymmetry and can be divided into two Gaussian peaks: one at 531.5 261 eV for the O2- assignment to the Zn-O bond, and another at 532.65 eV assigned to surface 262 hydroxyl oxygen (OH) due to structural defects. Please correct the position of O2- and OH in the spectra
Response: We are very grateful for your suggestions. According to the suggestions of reviewer, we have modified the corresponding positions in the paper.
12. For the XPS spectra of Ag (Figure 3b), no description was given in the discussion. Please explain the Ag peaks and significance
Response: We are very grateful for your suggestions. According to the suggestions of reviewer, we have modified the corresponding positions in the paper.
(A filling scheme has been added to the revised manuscript, lines 286-289 on the page 7.)
In the high-resolution XPS spectra of Ag 3d (Figure 5c), the binding energy peaks at 373.15 and 367.15 eV correspond to Ag 3d3/2 and Ag 3d5/2, respectively . The splitting energy between Ag 3d5/2 and Ag 3d3/2 is 6.0 eV , demonstrating that the Ag element in the composite is present in the Ag0 state.
13. What about the UV-visible absorbance of the fabricated substrates? Please include absorbance or reflectivity data
Response: Thank you for the comment. Your suggestion is very good, but due to the impact of the epidemic, our school is in a closed state and there is no relevant equipment on campus for testing, we will make improvements in the follow-up work.
14. The enhanced SERS activities of ZnO NPs@Ag NWs should be explained in terms of the charge transfer mechanism. The functional properties of ZnO should be explained in the introduction section.
Response: We are very grateful for your suggestions. According to the suggestions of reviewer, we have modified the corresponding positions in the paper.
(A filling scheme has been added to the revised manuscript, lines 68-77 on the page 2, lines 317-327 on the page 8.)
In addition, semiconductor-metal bonding has also been used to achieve highly sensi-tive SERS substrates, owing mainly to the charge transfer between the substrates and the probe molecule through the conduction band of the semiconductor, which is a chemical enhancement (CM)[48,49]. Semiconductors have an energy gap between the full valence band (VB) and the empty conduction band (CB), therefore, the charge transfer between semiconductor nanomaterials and molecules depends on the cou-pling between their energy levels (including CB, VB, highest occupied molecular orbital (HOMO) and lowest unoccupied molecular orbital (LUMO)) and their energy levels. As an environmental friend endly material, zinc oxide (ZnO) is recognized as an im-portant photocatalyst characterized by a hexagonal fibrillated zincite structure and a wide forbidden band of 3.37 eV[50,51]. The appropriate amount of ZnO loading acts as an electron transfer bridge between the silver Fermi energy level and the LUMO of the molecule, which is the key to play its important and bridging role in the CT effect.
The suitable ZnO NPs loading on the Ag NWs is significant to SERS performances. Firstly, we characterised the morphology of ZnO NPs@Ag NWs (0g, 0.2g, 0.02g, 0.005g) with SEM (Figure 4a-4d). Also, taking R6G as the molecular probe, the effect of the amount of ZnCl2 (0.2 g, 0.1 g, 0.05 g, 0.02 g, 0.01 g, 0.005 g) on the SERS intensity of the composites was investigated, the corresponding SERS spectra are shown in Figure 4e. We can see that the SERS performance of pure ZnO is far inferior to that of AgNWs, while the composite of ZnO and AgNWs significantly improves the SERS sensitivity. The CT enhancement mechanism can be explained by Figure S5, which shows (1) the charge transfer between the Fermi energy level of the molecule and the LUMO energy level of the molecule. (2) The appropriate amount of ZnO loading acts as an electron transfer bridge between the Fermi energy level of the silver and the LUMO of the mol-ecule, forming an effective "donor-bridge-acceptor" system. Therefore, the excellent surface-enhanced Raman scattering performance of silver/ZnO nanoparticles can be attributed to the synergistic CT effect of plasmonic silver, semiconductor ZnO and molecules.
15. Sample ID given in Figure 4e should be modified, especially the font size, so it will be readable. Also, modify Figure 2 (make the indexing to be readable)
Response: We are very grateful for your suggestions. According to the suggestions of reviewer, we have modified the corresponding positions in the paper.
